# Fast Explanation of RBF-Kernel SVM Models Using Activation Patterns

## Abstract

Machine learning models have significantly enriched the toolbox in the field of neuroimaging analysis. Among them, Support Vector Machines (SVM) have been one of the most popular models for supervised learning, but their use primarily relies on linear SVM models due to their explainability. Kernel SVM models are capable classifiers but more opaque. Recent advances in eXplainable AI (XAI) have developed several feature importance methods to address the explainability problem. However, noise variables can affect these explanations, making irrelevant variables regarded as important variables. This problem also appears in explaining linear models, which the linear pattern can address. This paper proposes a fast method to explain RBF kernel SVM globally by adopting the notion of linear pattern in kernel space. Our method can generate global explanations with low computational cost and is less affected by noise variables. We successfully evaluate our method on simulated and real MEG/EEG datasets.

## 1 Introduction

In neuroimaging data analysis, obtaining a high machine learning model performance is not the only objective. The ultimate goal of neuroscience is to gain insights into how information is passed and processed in the brain. Analyzing, interpreting, and visualizing the information that a model uses is crucial to this task. Enabling the explainability of a model's predictions is critical to fulfil this need.

The Support Vector Machine (SVM) is a widely used model in neuroimaging data analysis due to its simplicity, ability to handle outliers, and exceptional performance. Although linear SVM models are most commonly deployed, kernel-based SVM models can be utilized in nonlinear scenarios and extract more extensive information from the data. Compared with linear SVM models, with sufficient data, these models perform better in some neuroimaging analysis tasks (Sekkal et al., 2022). However, the use of these models is limited, one reason being their explainability issues. Due to the recent development of the XAI technique, a typical way to understand these models is using feature importance score (Carvalho et al., 2019). It should be noted that both linear and nonlinear model explanations may be influenced by class-irrelevant variables, such as suppressor variables (Krus & Wilkinson, 1986), which can inflate the importance scores of features that are, in actuality, irrelevant to the classification task at hand (Wilming et al., 2022).

In order to address this issue, Haufe et al. (2014) proposed a method which relies on covariance patterns. Assuming the observations can be generated from a generative model, the classifier weights can be transformed into an activation pattern that reflect the class-related features. However, this method is only applicable to linear models.

This notion can also be adopted when using RBF-SVM models. These models assume kernel functions can take data into higher dimensional space where data can be linearly separated. We show that the activation pattern can be constructed in kernel space, and the feature importance can be measured by mapping this high-dimensional pattern back into input space. This feature importance measure is less affected by noise variables like suppressor variables.

This paper proposes a novel explanation method for RBF-kernel SVM focusing on neuroimaging data analysis. The contributions are summarised as follows:

1. We propose a novel activation pattern based method for explaining RBF-kernel SVM in neuroimaging analysis tasks. We called this method Estimated Activation Pattern (EAP).

2. This method generates a global feature importance explanation and is more robust to noise variables than comparable approaches.

3. We show the empirical times of different methods. This method requires less computational cost.

This paper proceeds as follows. Section 2 introduces background information and related works. Section 3 presents our proposed method. Section 4 details the experiment evaluation and results. Section 5 concludes the paper with a discussion of further work.

## 2 BACKGROUND AND RELATED WORKS

Neuroimaging analysis usually serves two purposes: to successfully decode the signal, like brain-computer interface, or to infer knowledge of brain function by explaining the trained models. Mass univariate analysis (Groppe et al., 2011) is a common method that can help gain knowledge from neuroimaging data. These methods concentrate on single variables. Multivariate pattern analysis (MVPA) (Haxby, 2012), based on machine learning techniques, has become a popular alternative because of its ability to obtain multivariate patterns that consider multiple variables simultaneously.

To gain insight from trained models, one way is directly using the model performance as information, such as searchlight (Kriegeskorte et al., 2006), quantifying how much information a group of channels contains through model performance changes. Another way involves extracting knowledge from model structures, as demonstrated by the study on linear patterns (Haufe et al., 2014). The evolution of eXplainable Artificial Intelligence (XAI) has significantly enhanced the toolkit for comprehending learned information, particularly in dealing with complex models (Shi et al., 2020; Selvaraju et al., 2017; Kim et al., 2016; Goodfellow et al., 2014). As one of the most popular methods in neuroimaging analysis, efforts have been made to explain the non-linear SVM model. Barakat & Bradley (2010) have shown using rules as explanations. The problem with this method is that explanations may be hard to understand as the number of rules increases. Recently, several feature importance methods have been introduced in this field (Valentin et al., 2020; Bennett et al., 2021). However, some studies (Budding et al., 2021) report these methods would also be affected by noise variables even in linear models (Haufe et al., 2014).

## 3 METHODS

### 3.1 LINEAR ACTIVATION PATTERN

Explanation using model weights is of limited value because class-uncorrelated variables can have large weights. For example, large weights may be assigned to class-uncorrelated variables for better accuracy, like to fulfil the model structure need. In (Haufe et al., 2014), they addressed this problem and proposed a method to solve it in neuroimaging analysis tasks. In their work, the $\mathbf{n}$ d-dimension observations $\mathbf{X} \in \mathcal{R}^{n \times d}$ are assumed to be generated by k latent factors $\mathbf{L} \in \mathcal{R}^{n \times k}$ using specific patterns $\mathbf{W}_{generative} \in \mathcal{R}^{d \times k}$, which is: $\mathbf{X} = \mathbf{L}\,\mathbf{W}_{generative}^{T} + \epsilon$ , where the $\epsilon$ represents the noise variables. The latent variables could be a certain brain process or the different classes interested in the analysis task. It can be seen as an encoding process using pattern $\mathbf{W}_{generative}$. While the classier is a decoding process which is $\mathbf{L} = \mathbf{X}\,\mathbf{W}_{linear}$, where the $\mathbf{W}_{linear} \in \mathcal{R}^{d \times k}$ is the weights of the linear model.

By assuming the latent variables are independent, they conclude that with any classifier $\mathbf{L} = \mathbf{X}\,\mathbf{W}_{linear}$, the corresponding pattern can be constructed as $\mathbf{W}_{generative} = \Sigma_{\mathbf{X}}\mathbf{W}_{linear}\Sigma_{L}^{-1}$, where $\Sigma_{\mathbf{X}}$ represent the covariance matrix of observations $\mathbf{X}$ and $\Sigma_{L}$ is the covariance matrix of latent factors $\mathbf{L}$. Specifically for binary classification cases, i.e. $\mathbf{k} = 1$, since there is only one non-zero constant element in $\Sigma_{L}$, the above pattern can be simplified as $\mathbf{W}_{generative} \propto \Sigma_{\mathbf{X}}\mathbf{W}_{linear}$.

## 3.2 CONSTRUCT ACTIVATION PATTERN IN KERNEL SPACE

We can extend this idea and construct the pattern in kernel space. The SVM model, both linear and non-linear, aims to obtain a hyperplane between two classes. For the kernel-based SVM model, this hyperplane is represented by part of the data after mapped (assuming the unknown dimension is $\mathbf{m}$), the support vectors $\mathbf{S} \in \mathcal{R}^{s \times m}$ with associated coefficients $\alpha \in \mathcal{R}^s$, where the number of support vectors is $\mathbf{s}$. This model assumes that the mapping function $\phi(.)$ can map non-linearly separable data from input space into high-dimensional space where data can be separated with a hyperplane. This high-dimensional hyperplane can be represented as: $\mathbf{W}_{kernel} = \Sigma_i^s \alpha_i \, \phi(\mathbf{x}_i)$, where $\phi(\mathbf{x_i}) \in \mathcal{R}^m$ represent the vector in kernel space. To make it clear for the calculation shown below, we rewrite this into matrix format:

$$\mathbf{W}_{kernel} = \mathbf{S}^T \alpha \tag{1}$$

The mapped data covariance can also be represented in kernel space. There is no guarantee that the mapped data have zero means. For calculation convenience, we introduce the centering matrix $\mathbf{H} = \mathbf{I_n} - \frac{1}{\mathbf{n}}\mathbf{1_n}$, where $\mathbf{H} \in \mathcal{R}^{n \times n}$ and the $\mathbf{I_n} \in \mathcal{R}^{n \times n}$ is $\mathbf{n}$ dimensional identity matrix and $\mathbf{1_n}$ is the n-by-n matrix of all 1. $\mathbf{FH}$ can be seen as the sample minus mean step while calculating variance/covariance. The centering matrix has the useful property that $\mathbf{HH}^T = \mathbf{H}$, which can reduce calculation steps. The mapped covariance matrix is as follows:

$$
\begin{aligned}
\Sigma_{\phi(\mathbf{x})} &= \frac{1}{n}\mathbf{F}^T\mathbf{H}(\mathbf{F}^T\mathbf{H})^T \\
&= \frac{1}{n}\mathbf{F}^T\mathbf{H}\mathbf{F}
\end{aligned}
\tag{2}
$$

where $\mathbf{F} \in \mathcal{R}^{n \times m}$ and $\mathbf{F} = [\phi(\mathbf{x}_1), \phi(\mathbf{x}_2), ..., \phi(\mathbf{x}_n)]$ represent the mapped data matrix.

Combining equation 1 and 2, the pattern $\mathbf{W}_{\phi-generative} \in \mathcal{R}^m$ can be constructed in kernel space as:

$$
\begin{aligned}
\mathbf{W}_{\phi-generative} &= \frac{1}{n}\Sigma_{\phi(\mathbf{x})}\mathbf{W}_{kernel} \\
&= \frac{1}{n}\mathbf{F}^T\mathbf{H}\mathbf{F}\mathbf{S}^T\alpha
\end{aligned}
\tag{3}
$$

We should note that the current pattern $\mathbf{W}_{\phi-generative}$ is in high-dimensional space, which we do not know specifically. However, this is still a combination of mapped data $\mathbf{F}$. We can easily calculate the combination coefficient $\mathbf{P} \in \mathcal{R}^m$ as follows:

$$\text{Kernel Pattern Coefficient}: \ \mathbf{P} = \frac{1}{n}\mathbf{H}\mathbf{F}\mathbf{S}^T\alpha \tag{4}$$

## 3.3 MAPPING METHOD: FIXED POINT ITERATION

After constructing the pattern in kernel space, the next step is to map this result back to the input space. To address this problem, we can apply pre-image techniques(Honeine & Richard, 2011; Kwok & Tsang, 2004). Those methods have previously been used for kernel PCA denoising (Mika et al., 1998; Takahashi & Kurita, 2002).

The idea is to search the associated result $\mathbf{x}^* \in \mathcal{R}^d$ in input space by minimizing the mean squared error (MSE) between $\mathbf{x}^*$ and the target $\mathbf{W}_{\phi-generative}$ in kernel space. The mean squared distance is as follows:

$$
\begin{aligned}
\text{MSE}(\mathbf{x}^*) &= ||\mathbf{W}_{\phi-generative} - \phi(\mathbf{x}^*)||^2 \\
&= \mathbf{P}^T\mathbf{F}^T\mathbf{F}\mathbf{P} - \phi(\mathbf{x}^*)^T\mathbf{F}\mathbf{P} - \mathbf{P}^T\mathbf{F}^T\phi(\mathbf{x}^*) + \phi(\mathbf{x}^*)^T\phi(\mathbf{x}^*)
\end{aligned}
$$

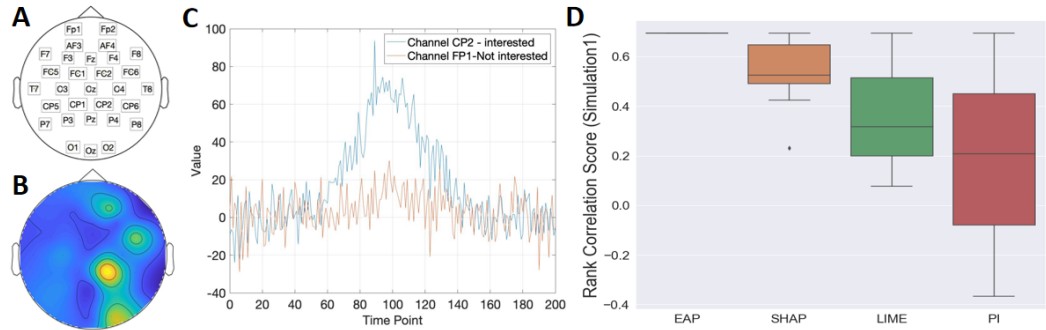

Figure 1: Example of EEG simulation dataset. Panel A shows the channel position of the simulation. Panel B shows the signal pattern of an example simulation dataset in experiment 1, and Panel C shows 2 example channels in this dataset. The light-colored area represents the class-related channel. These class-related channels are randomly selected. Panel D shows the rank correlation score between absolute explanation results and absolute generation weights.

where $\phi(\mathbf{x}^*) \in \mathcal{R}^m$ represents the mapped result $\mathbf{x}^*$ in kernel space. It is obvious that $\mathbf{P}^T\mathbf{F}^T\mathbf{FP}$ is a fixed constant. And $\phi(\mathbf{x}^*)^T \mathbf{FP} = \mathbf{P}^T\mathbf{F}^T\phi(\mathbf{x}^*) = \Sigma_i^n \mathbf{p}_i k(\mathbf{x}^*, \mathbf{x}_i)$, where $k(\mathbf{x}^*, \mathbf{x}_i)$ is the kernel function. After introducing the RBF kernel $k(\mathbf{x}_1, \mathbf{x}_2) = \exp\frac{||\mathbf{x}_1 - \mathbf{x}_2||^2}{\gamma}$, the minimizing function can be simplified as: $\arg\min_{\mathbf{x}^*} \mathrm{MSE}(\mathbf{x}^*) = k(\mathbf{x}^*, \mathbf{x}^*) - 2\Sigma_i^n \mathbf{p}_i k(\mathbf{x}^*, \mathbf{x}_i) = 1 - 2\Sigma_i^n \mathbf{p}_i \exp\frac{||\mathbf{x}_i - \mathbf{x}^*||^2}{\gamma}$

Here, we use the fixed-point iteration method to search the results. The fixed-point iteration method limited the search space, which decreased the computation cost. Furthermore, the result of this method will have the same scale as the input vectors have (Honeine & Richard, 2011), which makes the results more straightforward when using the pattern to explain the model. By setting the derivative for $\mathbf{x}^*$ to zero, the fixed-point iteration format is shown below:

$$\mathbf{x}^*_{t+1} = \frac{\Sigma_i^n \mathbf{p}_i k(\mathbf{x}_i, \mathbf{x}_t^*)\mathbf{x}_i}{\Sigma_i^n \mathbf{p}_i k(\mathbf{x}_i, \mathbf{x}_t^*)} \tag{5}$$

The fixed-point iteration method needs an initial vector for iteration. In our setting, all data should be scaled before training classifiers. So, we initialize the initial vector by sampling from a normal distribution with 0 mean and standard deviation of 10, i.e., using the same mean and larger variance than the scaled data. Some studies report that the fixed-point iteration method may suffer from instability and local minima problems (Abrahamsen & Hansen, 2009). Instead of directly using the final results, we log three solutions with the most minor loss score during the iteration to improve the stability. Then, calculate the normalization of the absolute values of these three solutions. The final solution will take the mean of these normalized results. Running multiple times with different initial vectors is recommended.

## 4 EXPERIMENTS AND RESULTS

This section conducts a series of experiments to evaluate our proposed method. The first two experiments use simulated electroencephalography (EEG) datasets. Experiment 3 uses a real visual task EEG and MEG dataset to evaluate how our proposed method performs in real-setting practice. The classifier implemented here is the RBF kernel SVM model. Three existing model explanation methods are implemented as comparisons.

### 4.1 DATASETS

#### 4.1.1 SIMULATION DATASET

Experiments 1 and 2 use ten simulated datasets each, all generated using the MVPA-Light MATLAB toolbox (Treder, 2020). These simulated EEG signals are structured as epoched Event-Related

Potential (ERP) settings, uniformly comprising 1000 samples categorized into two classes. Each sample is characterized by 30 channels and 200 time points, with a singular simulated peak signal.

---

**Algorithm 1:** Algorithm for EAP

**Input:**

     **N**:                 Number of iterations

     $\alpha, \gamma$:              Model coefficients of the trained SVM

     $[\mathbf{x}_1, \mathbf{x}_2, ..., \mathbf{x_s}]$: Support vectors of the trained SVM

     $[\mathbf{x}_1, \mathbf{x}_2, ..., \mathbf{x_n}]$: Training data

     **cth**:              The minimum changes

**Output:**

     $\mathbf{x}^*$:               Explanation vector

     `// Estimate pattern in kernel space`

1   Calculate centre matrix $\mathbf{H} = \mathbf{I}_n - \frac{1}{n}\mathbf{1}_n$ ;

2   **for** $i = 1{:}n$ **do**

3      **for** $j = 1{:}s$ **do**

4          Calculate the element $\mathbf{FS}^T = \mathbf{k}(\mathbf{x}_i, \mathbf{x}_j)$ at row **i** and column **j** using the kernel function with $\gamma$ used by the model.;

5          **j++** ;

6      **end**

7      **i++** ;

8   **end**

9   Calculate the coefficient of estimated pattern $\mathbf{P} = \frac{1}{n}\mathbf{HFS}^T\alpha$. The estimated pattern in kernel space is $\sum_{i=1}^{n} \mathbf{p}_i\phi(x_i)$.

     `// Mapping the estimated pattern into kernel space`

10   $\mathbf{t} \leftarrow 0$ ;

11   $\mathbf{x}_0^* \leftarrow$ initialised from standard normal distribution;

12   $\mathbf{diff} \leftarrow 1$;

13   **while** $(\mathbf{t} \leq \mathbf{N})$ *AND* $(|\mathbf{diff}| < \mathbf{cth})$ **do**

14      $numerator \leftarrow 0$;

15      $denominator \leftarrow 0$;

16      **for** $i = 1{:}n$ **do**

17          numerator = numerator + $\mathbf{p}_i\mathbf{k}(\mathbf{x}_i, \mathbf{x}_{t-1}^*)\mathbf{x}_i$ ;

18          denominator = denominator + $\mathbf{p}_i\mathbf{k}(\mathbf{x}_i, \mathbf{x}_{t-1}^*)$ ;

19          $\mathbf{i}++$;

20      **end**

21      $\mathbf{x}_t^* = \frac{numerator}{denominator}$ ;

22      $\mathbf{L}_t = \text{MSE}(\mathbf{x}_t^*)$;

23      **if** $\mathbf{t} > 1$ **then**

24          $\mathbf{diff} = \frac{\text{MSE}(\mathbf{x}_{t-1}^*) - \text{MSE}(\mathbf{x}_t^*)}{\text{MSE}(\mathbf{x}_{t-1}^*)}$

25      **end**

26      $\mathbf{t}++$;

         `// Log 3 different solutions with the smallest loss`

27      $\mathbf{x}_{l1}^*, \mathbf{x}_{l2}^*, \mathbf{x}_{l3}^* \leftarrow$ log three solutions with the most minor loss score during the iteration. ;

28   **end**

29   $\mathbf{x}_{l1}^*, \mathbf{x}_{l2}^*, \mathbf{x}_{l3}^* \leftarrow$ calculate the absolute value and normalised. ;

30   $\mathbf{x}^* = (\mathbf{x}_{l1}^* + \mathbf{x}_{l2}^* + \mathbf{x}_{l3}^*)/3$

---

The generation process involves creating a base signal through a normal probability density function, subsequently normalized with fixed mean and standard deviation across all samples. To introduce variability akin to real-world scenarios, each base signal is modulated by a random number drawn from the standard normal distribution. The signal matrix of each sample is generated by multiplying this modulated base signal with a pattern vector, representing signal strength across channels/features. Moreover, smoothed Gaussian noise is added to each channel. **signal**, **distractor** and **noise** are generated using the same strategy shown above while the pattern vectors differ. The

pattern vectors for **signal** and **distractor** are pre-defined and for **noise** are all zeros, i.e., pure noise. Figure 1-A shows the channel topography and data samples.

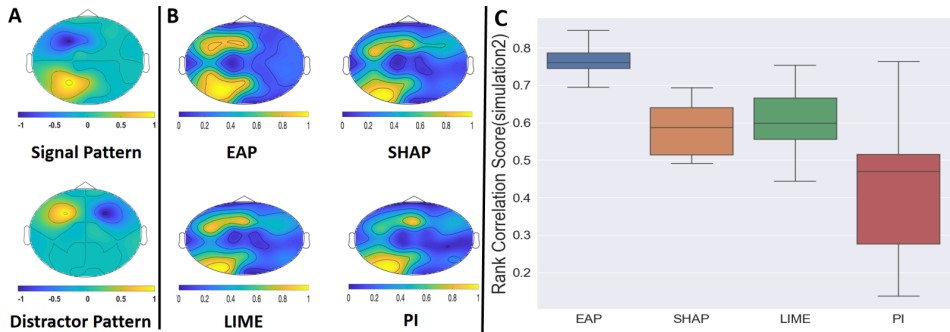

Figure 2: The patterns for the simulation dataset in experiment 2 and results. The upper plot of panel A shows the simulation generation weights of the signal pattern. The lower plot of panel A shows the simulation generation weights of the class-irrelevant distractor pattern. These distracting signals are irrelevant to the different classes but may affect the explanation results. Panel B shows the explanation results of different methods. Panel C shows the rank correlation score between the absolute weights of the signal pattern and absolute explanation results.

**Experiment 1** These datasets consist of two parts: **signal** and **noise**. For **signal**, the pattern vector is zero except for 6 randomly selected channels which assigned positive weights. The signal pattern can be seen as the ground truth. To introduce nonlinearity, this signal pattern undergoes multiplication by varying coefficients, producing two classes: positive ($\{1, -1\}$) and negative ($\{0.5, -0.5\}$). The composition of the simulation data is $\mathbf{X} = 0.25 \times \mathbf{signal} + 0.75 \times \mathbf{noise}$.

**Experiment 2** These datasets consist of three parts: **signal**, **distractor** and **noise**. The pattern vectors for **signal**, **distractor** are pre-defined which are shown in figure 2-A. Similar to Experiment 1, the signal pattern undergoes multiplication by distinct coefficients for positive and negative classes, while the distractor pattern is modulated by random numbers drawn from the standard normal distribution to mimic signal changes. The composition of the simulation data is $\mathbf{X} = 0.25 \times \mathbf{signal} + 0.25 \times \mathbf{distractor} + 0.5 \times \mathbf{noise}$.

### 4.1.2 EXPERIMENT 3: REAL DATASET

We used a visual task neuroimaging dataset (Wakeman & Henson, 2015). The EEG and MEG signals are measured using Elekta Neuromag Vectorview 306 system. All sixteen participants are asked to see pictures of faces and scrambled faces. For each participant, around 290 samples are logged for each class.

**Preprossessing** Those irrelevant channels, such as ECG and EOG, are first removed. Then applying bandpass filter between 1Hz to 40 Hz with windowed sinc Finite Impulse Response (FIR) filters. For EEG data, signals are re-referenced using the average reference method. After referencing, the data is downsampled to 220HZ to save computation costs. Then trials/samples are segmented based on the event file provided by the dataset. This step can ensure the event-related signal will appear at a similar time point for all trials. Each trial contains 0.5s before seeing the picture, and 1s after. Baseline correlation is applied based on a time window from 0.5s to 0s before the stimulus, which can highlight the signal differences. Finally, 70 EEG and 102 MEG channels (magnetometer) are selected for classification. All preprocessing tasks are carried out using Fieldtrip and MVPA-light toolbox on MATLAB.

Unlike the simulations in which the number and time of peak are fixed, the interested signal of real MEG/EEG data should be picked out first. Two interest time intervals are detected based on local minima in the Global Field Power (Skrandies, 1990). The two time intervals are 70 – 145 ms and 145 – 220 ms.

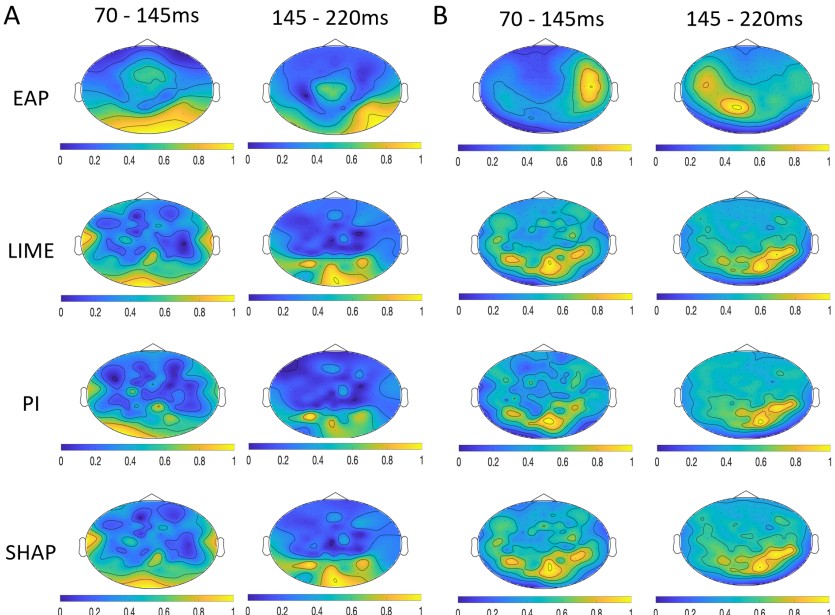

Figure 3: The results of experiment 3. Figure A shows the explanation results of EEG data. Figure B shows the explanation results of MEG data.

## 4.2 CLASSIFIER

In all three experiments, we use the statistical feature of the signal (Blankertz et al., 2011), i.e., the mean channel value among selected time intervals, as the classification feature. All classifiers are built in Python using Sci-kit learn toolbox. Hyperparameters are tunning based on 5-fold cross-validation. Before building models, data are first scaled using the standard scaler as default.

## 4.3 COMPARISON METHODS

Several state-of-the-art explanation methods are selected for comparison, including permutation importance (PI) (Fisher et al., 2019; Breiman, 2001), local interpretable model-agnostic explanation (LIME) (Ribeiro et al., 2016), and Shapley additive explanations (SHAP) (Lundberg & Lee, 2017).

These methods will generate results in different scales. The explanations may not have the same scale. For the convenience of comparison, all results use their absolute values and then normalized between 0 to 1.

## 4.4 RESULTS

### 4.4.1 EXPERIMENT 1

Figure 1 shows the results of experiment 1. As mentioned in the previous section, the randomly assigned signal pattern can be seen as the ground truth of experiment 1. The rank correlation score is calculated to show how the explanation results performed. As shown in figure 1, all methods can obtain good results in experiment 1. The correlation score of EAP outperformed other methods and showed a low variance compared to other methods. The permutation importance shows the worst results.

### 4.4.2 EXPERIMENT 2

The results shown in figure 2-B are the averaged results of simulation datasets. All simulation datasets have the same ground truth. Under the view of averaged results, all methods have roughly picked up the true area shown in the signal pattern. While the EAP result shows less affected by

Table 1: Average run time (in seconds) of all Methods.

|  | PI | LIME | SHAP | EAP |
|---|---|---|---|---|
| Experiment 1 | 4.03 | 554.44 | 18414.00 | 2.24 |
| Experiment 2 | 4.43 | 573.18 | 18730.36 | 2.45 |
| EEG $1^{st}$ interval | 10.93 | 752.26 | 11763.12 | 2.30 |
| EEG $2^{nd}$ interval | 9.18 | 8984.92 | 9060.15 | 1.88 |
| MEG $1^{st}$ interval | 9.55 | 717.92 | 10401.64 | 1.83 |
| MEG $2^{nd}$ interval | 13.15 | 12531.63 | 9828.30 | 1.53 |
| Iris-versicolor-vs-virginica | 0.02 | 3.33 | 42.95 | 0.07 |

the distractor signal, the other three methods are, to some extent, affected by the distractor factors. Figure 2 Panel C shows the rank correlation score between the absolute signal pattern and the explanation results. Like the results shown in experiment 1, the EAP methods outperformed the other three methods with lower variance. The results have indicated that the EAP method can generate robust explanation results under the effect of distractor factors.

### 4.4.3 EXPERIMENT 3

Figure 3 shows the results of experiment 3. All explanation results are taken in absolute value, rescaled to 0 to 1, and averaged over the 16 participants. Unlike simulation datasets, we do not have the exact ground truth of real-world MEG/EEG data. But we can compare our results with previous studies.

The selected time intervals are consistent with the two components of the visual tasks brain cognitive process, which are P100 and N170. These two components have been reported in many previous studies (Boutros et al., 1997; Kropotov, 2016).

As shown in the figure, all methods in EEG experiments in the first time interval (70-145ms) highlight the bottom back area, which reflects the occipital area. PI, LIME, and SHAP also highlight the two side areas of both hemispheres, which reflect the temporal area of the brain. In the second time interval (145-220ms), the EAP method mainly focuses on channels located at the occipital and occipital-temporal areas on the right hemisphere. While the other 3 methods focus mainly on channels located in the middle occipital area. The first time interval reflects the P100 component in visual-related studies. Channels located in the middle occipital area are reported in previous studies (Negrini et al., 2017; Rossion & Jacques, 2008). These channels show signal differences related to face vs. non-face stimulus around 100ms after the stimulus. The second time interval reflects the N170 component. Signal differences are reported at channels located at the occipital-temporal area on both hemispheres Negrini et al. (2017); Rossion & Jacques (2008). In contrast, channels located in the right occipital-temporal area detect significantly stronger signals than the left counterpart (Wang et al., 2019; Henson et al., 2009). In summary, compared with all methods, the EAP method better reflects the previous study.

For results of MEG data, in the first time interval, the EAP method highlights the channels located in the anterior area of the right hemisphere, which is the temporal area. While other methods highlight channels located in occipital and occipital-temporal areas. In the second time interval, the EAP result highlights the left occipital-temporal area. The other 3 methods highlight the middle occipital area and right occipital-temporal areas. In previous studies, in the early stage ( around 100 ms after stimulus), some report almost no channel level difference between face vs. non-face stimulus (Liu et al., 2002). In contrast, channel level differences are found at channels located at the right temporal area, left frontal-temporal area, and right occipital-temporal area at around 150 ms(Tadel et al., 2019). For the second time interval, channel level differences are reported at the occipital-temporal area of both hemispheres Liu et al. (2002); Xu et al. (2005). However, channel level differences are reported at the left occipital-temporal area and the right temporal area at around 200 ms. In summary, all methods partly reflect the previous studies.

Table 2: Normalised feature importance score (iris-versicolor vs iris-virginica)

|  | Sepal length | Sepal width | Petal length | Petal width |
|---|---|---|---|---|
| **PI** | 0.0376 | 0 | 0.9037 | 1 |
| **LIME** | 0.0785 | 0 | 0.6554 | 1 |
| **SHAP** | 0.0666 | 0 | 0.6189 | 1 |
| **EAP** | 0.5028 | 0 | 1 | 0.9775 |
| P-value(T-test) | 1.72E-07 | 0.0018 | 3.18E-22 | 2.23E-26 |

#### 4.4.4 EMPIRICAL COMPUTATION COST

This subsection provides an empirical comparison of computational cost between the proposed method and other benchmark methods. The experiment is carried out on a desktop with i7 9700k CPU with 32 GB ram. The operating system is Ubuntu 20.04. Table 1 summarizes the run time, which shows our EAP method is multiple order faster than LIME and SHAP and a few times faster than PI.

#### 4.4.5 ADDITIONAL EXPERIMENT ON NON-NEUROIMAGING DATA

The proposed method primarily focuses on analyzing neuroimaging data. However, we are also interested in exploring the applicability of our methodology to handle general tabular data. To this end, we evaluate the proposed method using the iris dataset with the Versicolor class and the Virginia class. Figure 4 shows the feature distribution between iris-versicolor and iris-virginica. We also perform t-test between the two classes for each feature as listed in table 2. Based on visual observation of Figure 4 and the t-test results in table 2, petal length and petal width are the most useful features and sepal width is the least useful, while sepal length is medially useful. The experimental results in table 2 indicate that all methods correctly highly rate the two most important features of petal length and petal width and dismiss the least important feature of sepal width. However, the EAP method can better identify the usefulness of sepal length while other explanation methods underestimated the importance score of this feature as shown in table 2.

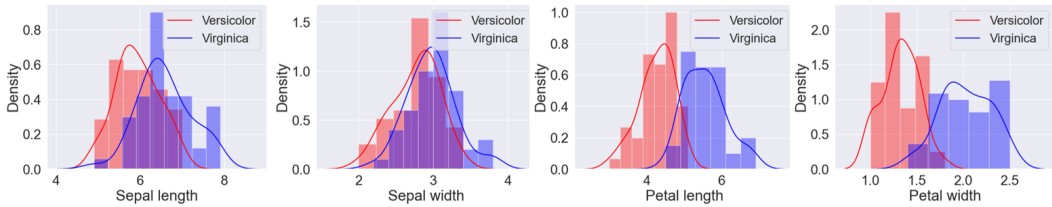

Figure 4: The feature distribution of the features in the iris dataset between iris-versicolor and iris-virginica. From the plot, the Sepal length can provide some information to distinguish the two classes, while the sepal width cannot.

## 5 CONCLUSION AND FURTHER WORK

In summary, this paper proposes a novel explanation method EAP for the RBF kernel SVM model. We implement our method and three other state-of-art explanation methods to the EEG simulation dataset and real MEG/EEG dataset of visual tasks. The experimental results show that the EAP can obtain the global explanation results with relatively low computation costs. Furthermore, the EAP method can generate robust explanation results and is less affected by distractor factors.

This method may also have potential for other kinds of kernel types, which is one of our future directions. A limitation of our method is that the EAP method currently can only generate global explanations. A fruitful direction would be to extend our method to generate local explanations. Another future direction is applying our method to other kinds of kernel methods. These future directions would greatly enlarge the flexibility of our proposed method.

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

## A  APPENDIX

You may include other additional sections here.

