# OpenReview forum: "Fast Explanation of RBF-Kernel SVM Models Using Activation Patterns"
_ICLR.cc/2024/Conference — ICLR 2024 Conference Withdrawn Submission_

### Official Review · Reviewer_bpWC · 2023-10-27

**Soundness:** 4 excellent
**Presentation:** 4 excellent
**Contribution:** 4 excellent
**Rating:** 8
**Confidence:** 4

**Summary:**

This paper extends an existing method for explanation of linear models to RBF kernels. The authors show how to extract approximate explanations from this extension by minimising distances in the embedded space. Experiments on simulated and real data indicate that these explanations are substantially better than some existing state-of-the art generic explanation methods.

**Strengths:**

Interpretation of non-linear SVMs is a important and impactful problem, particularly in the science where the focus may be more on discovery than performance. This method is straight forward, efficient, and seemingly very effective.

**Weaknesses:**

There is no exploration of how difficult the problem of extracting a pattern through minimising the MSE is. It's stated that the optimisation is run three times, with the mean being used to summarise them. It would be interesting to explore the stability of the iteration more, as this is the critical part of the algorithm. It is also unclear if the mean is the best summary statistic as there may be multiple equally satisfactory explanations possible.

Only neuroimaging data is considered, both for simulations and the real world dataset. It would be interesting to know how this method scales with dimensionality, and if it still remains useful in p >> n scenarios.

**Questions:**

- Have high dimensional datasets been investigated?
- It would be great to understand more the stability of the MSE optimisation, and if the mean is appropriate.

---

> ### Author Response · Authors · 2023-11-22
>
> Thanks for your suggestions. The questions are very similar to weakness, and we answer them together below.
>
> Question1: Have high dimensional datasets been investigated?
> Answer:
> Thank you for your suggestion regarding investigating the performance of our method with dimensionality and sparse small datasets. Your suggestions provide us with additional opportunities to enhance our approach.  We are currently working on testing our method with various real-world datasets, especially those involving tabular data. Given the tight rebuttal deadline, we managed to add an example of iris data to illustrate that our method can be used to explain the feature importance of tabular data. Although we have not yet fully completed this task, we will definitely consider incorporating this kind of dataset into our future work, following your suggestion and idea. Thank you again for your valuable suggestion.
> Furthermore, another problem that may arise when using a very high dimensional dataset, is that other benchmark methods are needed since our current benchmark explanation methods (Permutation importance, SHAP and LIME) all suffer from the dimensional increase.  As the dimension increases, the runtime of these methods becomes impractical to experiment.
>
> Question2: It would be great to understand more the stability of the MSE optimisation, and if the mean is appropriate.
> Answer:
> Thank you for your valuable comments and suggestions that help us improve our work.
> We agree that the optimization process can be improved, and we are currently working hard to enhance our optimization method. Our efforts are focused on developing methods that can be adapted to different kernel types. However, we are not done yet.  Our current optimization method is based on the fix-point iteration method, which has been previously utilized to solve pre-image problems. One of the drawbacks of this method is that it may encounter local minima issues. While averaging the results may lead to less precise outcomes, in our specific cases, it could improve the stability. We have run our method on the simulation dataset several hundreds of times, far more than the results included in the manuscript. And the results are very stable.

---

### Official Review · Reviewer_KvcN · 2023-11-01

**Soundness:** 2 fair
**Presentation:** 1 poor
**Contribution:** 2 fair
**Rating:** 5
**Confidence:** 4

**Summary:**

Needs more work.

The writing is really not presenting the paper under the most favorable light. I think that should this paper be accepted, which I do not recommend, an extensive rewrite is absolutely needed for the sake of the readers and the conference attendees.
There are too many vague statements and too many unclear sentences that make reading the paper too cumbersome.

The subject is narrow, the theoretical contribution is not exceptional, the experimental results are too terse.
In particular, the experiments are not varied enough to validate the method in general.

**Strengths:**

1) Some experimental results are encouraging, in particular Figure 2C and the fact that the method proposed by the authors is much faster than the presented competitors.

2) The method is relatively simple and well documented in Algorithm 1.

**Weaknesses:**

1) The impact is too narrow in my opinion and the technique not innovative enough.

2) The experimental results are not diverse enough to really argue that the method is recommendable for general purpose XAI. The authors might want to have a deeper discussion about other kernels.

3) The writing of the paper is too loose which is detrimental to the reader and the conference's audience. E.g. while XAI is a "hot topic" I would not use that language in an academic paper, specifically a paper with a very narrow focus application-wise.

**Questions:**

1) Could the authors please expand the experimental section to make it more convincing to the general XAI audience?

2) Could the authors please deepen the discussion on other kernels and why they are not treated in the paper?

Thank you.

---

> ### Author Response · Authors · 2023-11-22
>
> Weakness1:
>
> Answer:
> We appreciate your perspective on the perceived narrowness of the impact. Our proposed method adapted the idea of linear pattern and expanded to RBF-SVM model. We are continuously striving to improve the scope of our work. This method has the potential to be applied to SVM models using other types of kernels.  We are currently making efforts to investigate these methods for other types of kernels and find better optimization methods. We have produced encouraging preliminary findings that suggest the potential usefulness of our methodology with other kernel types. Nevertheless, further investigations are still necessary. Furthermore, we are also trying to broaden this method to other kernel based methods, and this will be included in our future work.
>
> Weakness2:
>
> Answer:
> Thank you for providing your insightful feedback regarding the diversity of experimental results. We have demonstrated the usefulness of our methods for the ERP-like dataset, but we agree that it's important to showcase a wider range of experiments to strengthen the argument for the method's applicability in general-purpose eXplainable Artificial Intelligence (XAI). Currently, we are making efforts to investigate the potential of our method applied to different conventional tabular datasets. Given the tight rebuttal deadline, we managed to add an example of iris data to illustrate that our method can be used to explain feature importance of tabular data. , but unfortunately, we haven't finished yet, especially before the deadline.
> We appreciate your suggestion to explore a deeper discussion on alternative kernels, and it aligns with our commitment to comprehensive exploration. We are currently investigating the usefulness of this method on other kinds of kernels, and our initial findings suggest that this methodology can be extended to other kernel-based SVM models, albeit with ongoing research and analysis. We'll consider incorporating a more extensive analysis of various kernels to enhance the method's versatility and broaden its applicability.
>
> Weakness3:
> Answer:
> We apologize for any inconvenience caused by the writing quality. Despite time constraints, we are trying our best to improve the text by replacing vague phrases and correcting grammatical errors. Furthermore, we also address the writing issue by revising the language throughout the paper to ensure that it aligns more appropriately with the scholarly expectations of the conference audience. We aim to enhance the precision and clarity of our writing, ensuring that it reflects the focused and specialized nature of our research without compromising on academic rigour. We thank you for bringing this matter to our attention, and we are committed to improving the quality of our manuscript based on your valuable suggestions.
>
>
>
> Question1:
>
> Answer:
> We have making effort to find suitable more general tabular datasets to evaluate explanation methods. Unfortunately, we have not finish within the time limit.  Nevertheless, we adapted our method to commonly used tabular datasets, such as iris, and built an RBF-SVM classifier between Versicolor and Virginica, which are believed to contain nonlinearly separable information. The t-test indicates that the mean of sepal length between the two classes is significantly different. This indicates that this feature could potentially provide important information for classification, our method can identify the usefulness of sepal length while other explanation methods underestimated the importance score of this feature as shown in the table below.
> |                     | **Sepal length** | **Sepal width** | **Petal length** | **Petal width** |
> |---------------------|-----------------:|----------------:|-----------------:|----------------:|
> | **PI**              |      0.0376      |       0.0       |      0.9037      |       1.0       |
> | **LIME**            |      0.0785      |        0        |      0.6554      |        1        |
> | **SHAP**            |      0.0666      |        0        |      0.6189      |        1        |
> | **EAP**             |    **0.5028**    |        0        |         1        |      0.9775     |
> | **P-value(T-test)** |     1.72E-07     |      0.0018     |     3.18E-22     |     2.23E-26    |
>
> Question2:
>
> Answer:
> This method has the potential to be adapted to SVM models using other types of kernels. We are actively pursuing this direction and seeking to improve our optimization techniques. However, we have not finished yet, especially within the time limitation. We have produced encouraging preliminary findings that suggest the potential usefulness of our methodology with other kernel types. Nevertheless, further investigations are necessary. Our upcoming efforts will center on the expansion of this concept and the development of methods that can be applied to a broader range of kernels. We included this point in the updated paper when we discussed further work.

---

### Official Review · Reviewer_vJrk · 2023-11-02

**Soundness:** 3 good
**Presentation:** 2 fair
**Contribution:** 3 good
**Rating:** 6
**Confidence:** 4

**Summary:**

This paper concerns explainability in kernel methods, such as SVMs with RBF kernels.
The important problem of correlated noise typically leads to a confounded importance "heatmap".
Haufe et al. (cited in the paper) solved the linear case, while the non-linear case has been looking for good ideas.  Here the authors propose to use the approximate linearity of latent space in kernel methods to de-noise the explanation using Haufe et al.'s method there.

Simulation and benchmark cases are studied. In the simulation study where feature space ground truth is available, the proposed method outperforms relevant baselines.
In benchmark data where earlier analyses can be used as a proxy for ground truth, performance looks promising

The mathematical and algorithmic work is basic based on a straightforward application of kernel denoising by pre-imaging.

**Strengths:**

Simple and productive idea.
Straightforward development of an algorithm with some additional minor heuristics (explained in the algorithm 1 box).
Experimental results (including error estimates for the simulation study) supports the proposal.
Computational cost is much less than baseline methods.

**Weaknesses:**

Much of the math is so basic that a simple reference would have been enough. The gained additional space could have been used to explore the role and significance of the assumptions and heuristics used to stabilize the solution.

The writing is not always so clear - could have benefitted from a writing assistant.

**Questions:**

Please address the role and significance of the assumptions (linearity of latent space) and heuristics (ensemble averaging).

Consider running the manuscript with a writing assistant.

---

> ### Author Response · Authors · 2023-11-22
>
> **Weakness:** Much of the math is so basic that a simple reference would have been enough. The gained additional space could have been used to explore the role and significance of the assumptions and heuristics used to stabilize the solution.
> The writing is not always so clear - could have benefitted from a writing assistant.
>
> Answer:
>  We regret any inconvenience caused by the quality of the presentation. We acknowledge the reviewer's critique that the mathematical content may have been elementary, and while we have made some modifications, we have chosen to retain the majority of the math part as it is to make the paper self-contained for the convenience of the readers who need it. Despite the time constraints, we have endeavoured to improve the text by clarifying vague phrases and addressing grammatical inaccuracies. We trust these edits have contributed to a more lucid and accessible paper. We will improve further for the camera-ready submission.
>
>
>
>
> **Question: **Please address the role and significance of the assumptions (linearity of latent space) and heuristics (ensemble averaging)
> Consider running the manuscript with a writing assistant.
>
> Answer:
> Our approach relies on the assumption of the radial basis function (RBF) kernel. This kernel can map the current data into a kernel space of infinite dimensions, enabling the data to be separated linearly in that space. We can estimate the pattern in this kernel space.  Additionally, we assume that the observations (X) are produced by a latent source (target variable), enabling the estimation of observation generation weights (pattern) in binary classification scenarios where linear separability is present. This pattern can offer a more comprehensive understanding of feature significance.
> To obtain the input space counterpart of pattern results in kernel space, we introduce the fixed-point iteration method, which has been previously utilized to solve pre-image problems. However, this method may encounter local minima issues. While averaging the results may lead to less precise outcomes, in our specific cases, it could improve the stability.
> Sorry for the inconvenience of the writing quality. We have proofread the paper for the updated PDF of our paper that has now been uploaded to the review site, we hope you will find the linguistic quality of this version improved. We will continue improving the writing with additional help and have a much-polished manuscript for the camera-ready submission.

---

> > ### Comment · Reviewer_vJrk · 2023-11-23
> > **Accept the changes and maintain evaluation**
> >
> > I acknowledge the feedback from the authors and maintain the evaluation (6)

---

### Official Review · Reviewer_pU9a · 2023-11-03

**Soundness:** 3 good
**Presentation:** 3 good
**Contribution:** 3 good
**Rating:** 5
**Confidence:** 3

**Summary:**

This paper proposes a new explainer of the kernel SVM under the framework of interpretable machine learning or XAI, which is an active research area. Haufe et al (2014) proposed an explainer for linear models based on a latent variable approach with applications in neuroimaging analysis. The approach is extended to kernel SVM in this paper. As shown in simulations and real applications, the proposed method gives more cogent explanations but also runs faster when compared with the popular explainers such as SHAP and LIME.

**Strengths:**

The paper is working on a significant problem. The contribution is significant due to the improvement over the state-of-the-art explainers like LIME and SHAP.

**Weaknesses:**

1. The "Experiment" section requires more clarity. Simply mentioning that data generation utilizes a MATLAB toolbox isn't comprehensive. There's a need for details of the signal and noise pattern creation processes. It's important to highlight that the true data-generating model isn't linear, which underscores the preference for kernel SVM over its linear counterpart.
2. While the emphasis on neuroimaging data is pertinent, its scope appears restricted. The proposed technique seems to be quite general. Could it be naturally integrated with other kernel methodologies, like kernel logistic regression, for instance? The support vectors in SVM might be instrumental in expediting explanation computation; however, this strategy might be equally effective for kernel logistic regression. Furthermore, adapting this method for conventional tabular data could be valuable. In such a scenario, the resulting explanation vector could serve as a metric for variable importance.

**Questions:**

In addition to the queries outlined in the "Weakness" section, there is another question. Is it true that the proposed method roughly gives the same result with a naive approach which simply gives the mapped data $\phi(x)$, say by eigendecomposition, and directly applies the linear explainer proposed in Haufe et al (2014)? Even if the current proposal operates without the precise mapping of $\phi(x)$, might we obtain a comparable outcome if $\phi$ is discerned through a more rudimentary approach?

---

> ### Author Response · Authors · 2023-11-22
>
> Weakness1.
>
> Answer: We apologize for the confusion caused by manuscript writing quality. To generate a sample, the toolbox first creates a base signal using a probability density function with predefined mean and standard deviation. This signal is then multiplied by a random number to simulate sample differences. The base signal is then expanded into a multidimensional matrix by multiplying it with the signal pattern, which is the weights of the different channels or features. Finally, smoothed Gaussian noise is added to each channel or feature.  We have revised the draft in section 4.1.1 to introduce more details about the sample-generating function of the used toolbox. Hope this will be helpful.
> By adjusting the signal pattern, the weights of different channels can be modified, samples are much different between classes. Additionally, to ensure that the data is in linear inseparable cases, the true signal pattern is multiplied by coefficients for the two classes, positive (1, -1)  and negative (0.5, -0.5).  We hope this explanation provides more clarity on the generating functions used in the toolbox.
>
> Weakness2:
>
> Answer: Thank you for your valuable suggestions. We are currently working on adapting our method to other kernel-based SVM models, as well as implementing better optimization methods. The idea of adapting this method to other kernel-based methods is very attractive, and we have mentioned it in the draft.
> We applied our methods to kernel logistic regression(KLR). For the convenience of applying our code, we built the model based on sklearn Logistic Regression with the Nystroem method to approximate the kernel. We have adopted this method for the simulation data of experiment 2 used in the manuscript. The Nystroem method employs all training samples, yet recent results are not convincing: with a mean ranked correlation score of only 0.6355 and a standard deviation of 0.29. This differs notably from the results achieved through the RBF-SVM model. In the manuscript, the scaled set is used to estimate the covariance matrix in kernel space during training. However, utilizing unscaled data, which we think is too much information outside the model, for this process can enhance performance for both KLR and RBF-SVM models. Additionally, using unscaled data to calculate the covariance matrix can improve results in the linear kernel as well. One potential reason is that in KLR, all or most of the data are used to estimate distances in kernel space while in SVM, only samples influenced the decisions will be used. And this increased the complexity of optimizing the results. We agree that adapting this idea to other kernel methods is very attractive. Although the current results of KLR are not very satisfactory, this method could be improved by using other optimization methods which we will investigate further as a future work.
> As for adapting this method for conventional tabular data, due to the limited time, we only managed to adapte our method to commonly used tabular datasets, we added iris data as an illustration example and built an RBF-SVM classifier between Versicolor and Virginica, which assumes these two classes are nonlinearly separable. The t-test indicates that the mean of sepal length between the two classes is significantly different. This indicates that this feature could potentially provide important information for classification, our method can identify the usefulness of sepal length while other explanation methods underestimated the importance score of this feature as shown in the table below.
> |                     | **Sepal length** | **Sepal width** | **Petal length** | **Petal width** |
> |---------------------|-----------------:|----------------:|-----------------:|----------------:|
> | **PI**              |      0.0376      |       0.0       |      0.9037      |       1.0       |
> | **LIME**            |      0.0785      |        0        |      0.6554      |        1        |
> | **SHAP**            |      0.0666      |        0        |      0.6189      |        1        |
> | **EAP**             |    **0.5028**    |        0        |         1        |      0.9775     |
> | **P-value(T-test)** |     1.72E-07     |      0.0018     |     3.18E-22     |     2.23E-26    |
>
> Question1:
>
> Answer:
> Thanks for the questions. If We comprehend your question accurately, by assuming the mapped data are linearly separable, we can apply the linear pattern (Haufe et al, 2014). However, the obtained results will be the contribution of the mapped data features rather than the input feature.
> We think it might be possible to obtain more precise results if we can discern the $\phi$. Although the precise results could maintain a similar scale as the data in the training set, it can be hard to compare the results between different models. However, if we can discern $\phi$ precisely, it provides the possibility to directly compare the outcome in the mapped space, e.g., compare the similarity in mapped space.

---

### Author Response · Authors · 2023-11-22

We would like to express our sincere gratitude to the anonymous referees for their insightful comments and valuable suggestions on our paper. After careful consideration, we have addressed all the comments, implemented all the suggestions and made the necessary revisions to the manuscript. We have also included a new experiment to test the effectiveness of our method on a tabular dataset. The detailed response to each comment of individual reviewers is provided in the response area. We invite you to take a look at the updated pdf file of our paper, which we believe has significantly improved. Thank you for your time and consideration.